# Mesenchymal Stem Cell-Based Therapies for Temporomandibular Joint Repair: A Systematic Review of Preclinical Studies

**DOI:** 10.3390/cells13110990

**Published:** 2024-06-06

**Authors:** Yuanyuan Jiang, Jiajun Shi, Wenjun Di, Kristeen Ye Wen Teo, Wei Seong Toh

**Affiliations:** 1Faculty of Dentistry, National University of Singapore, 9 Lower Kent Ridge Road, Singapore 119085, Singapore; 2Center for Cleft Lip and Palate Treatment, Plastic Surgery Hospital, Chinese Academy of Medical Sciences and Peking Union Medical College, 33 Badachu Road, Shijingshan District, Beijing 100144, China; 3Department of Orthopaedic Surgery, Yong Loo Lin School of Medicine, National University of Singapore, 1E Kent Ridge Road, Singapore 119228, Singapore; 4Tissue Engineering Program, Life Sciences Institute, National University of Singapore, 27 Medical Drive, Singapore 117510, Singapore; 5Department of Biomedical Engineering, College of Design and Engineering, National University of Singapore, 4 Engineering Drive 3, Singapore 117583, Singapore; 6Integrative Sciences and Engineering Program, NUS Graduate School, National University of Singapore, 21 Lower Kent Ridge Road, Singapore 119077, Singapore

**Keywords:** mesenchymal stem/stromal cells, secretome, extracellular vesicles, exosomes, cartilage, temporomandibular joint, osteoarthritis, systematic review

## Abstract

Temporomandibular disorders (TMDs) are a heterogeneous group of musculoskeletal and neuromuscular conditions involving the temporomandibular joint (TMJ), masticatory muscles, and associated structures. Mesenchymal stromal/stem cells (MSCs) have emerged as a promising therapy for TMJ repair. This systematic review aims to consolidate findings from the preclinical animal studies evaluating MSC-based therapies, including MSCs, their secretome, and extracellular vesicles (EVs), for the treatment of TMJ cartilage/osteochondral defects and osteoarthritis (OA). Following the PRISMA guidelines, PubMed, Embase, Scopus, and Cochrane Library databases were searched for relevant studies. A total of 23 studies involving 125 *mice*, 149 *rats*, 470 *rabbits*, and 74 *goats* were identified. Compliance with the ARRIVE guidelines was evaluated for quality assessment, while the SYRCLE risk of bias tool was used to assess the risk of bias for the studies. Generally, MSC-based therapies demonstrated efficacy in TMJ repair across animal models of TMJ defects and OA. In most studies, animals treated with MSCs, their derived secretome, or EVs displayed improved morphological, histological, molecular, and behavioral pain outcomes, coupled with positive effects on cellular proliferation, migration, and matrix synthesis, as well as immunomodulation. However, unclear risk in bias and incomplete reporting highlight the need for standardized outcome measurements and reporting in future investigations.

## 1. Introduction

The temporomandibular joint (TMJ) is a ginglymoarthrodial (hinging and gliding) joint that connects the mandibular condyle with the temporal articular surface and is one of the most frequently used joints in the human body. Many pathological stimuli such as injury/trauma, joint overload, malocclusion, stress, and psychiatric illness can impact the joint structure and function, potentially leading to temporomandibular disorders (TMDs), a heterogeneous group of musculoskeletal and neuromuscular conditions involving the TMJ, masticatory muscles, and associated structures [1]. Among these conditions, osteoarthritis (OA) of the TMJ is an important subtype of TMDs and is characterized by synovial inflammation, disc degeneration, cartilage degradation, and subchondral bone erosion [2]. Patients with TMJ-OA suffer joint pain and dysfunction with a reduced quality of life. Based on clinical and magnetic resonance imaging (MRI) examination, the prevalence of TMJ-OA was found to be approximately 25% in the 20–49 age group, affecting more females than males [3].

There is presently no curative treatment for TMJ-OA, and current management is focused on conservative and non-invasive modalities, including physical therapies, occlusal splints/orthotics, non-steroidal anti-inflammatory drugs (NSAIDs), arthrocentesis, and intra-articular injections of hyaluronic acid (HA) [1,2]. Despite providing symptomatic relief to some extent, these treatments are unable to repair and restore the damaged cartilage and subchondral bone. Thus, restoring the integrity and function of the joint tissues, including cartilage and subchondral bone, is critical in halting or reversing the OA progression, where the final treatment option is prosthetic replacement [1]. However, repair and regeneration of the joint tissues, in particular the damaged cartilage, have been challenging as the cartilage generally has a poor healing capacity due primarily to it being a poorly vascularized, aneural, and alymphatic load-bearing tissue overlying the supporting subchondral bone.

In recent years, stem cells, particularly mesenchymal stem/stromal cells (MSCs), have emerged as a promising cell source for TMJ repair [4]. Residing as multipotent cells within several adult tissues, MSCs have been isolated from diverse tissues, including bone marrow, adipose tissue, skeletal muscle, synovium, dental tissues, peripheral blood, and others [5]. They can replicate as undifferentiated cells and have the potential to differentiate into multiple lineages of the mesenchymal tissues, including bone, cartilage, fat, and muscle [5].

MSCs have been reported to be safe and effective for articular cartilage repair of the knee, with the added benefits of lower cost and lesser donor site morbidity [6]. Other clinical studies have also reported the efficacy of intra-articular MSC injections to improve pain and functions for knee OA, as recently reviewed [7]. In the context of TMJ repair, intra-articular injections of autologous bone marrow nucleated cells [8] or fat-derived stem cells containing MSCs [9] reported improved clinical outcomes in patients with TMJ-OA or derangement. Despite their therapeutic potential, the clinical translation of such an approach using MSCs remains somewhat limited.

Although MSCs were thought to act therapeutically as stem cells via cellular differentiation and replacement, it is now accepted that these cells mediate tissue repair through their paracrine secretion and may be more appropriately termed “medicinal signaling cells” [10]. The secreted factors, collectively known as the secretome, are composed of soluble proteins, free nucleic acids and lipids, and extracellular vesicles (EVs). Being a heterogeneous class of lipid membrane vesicles released by cells into the extracellular space, EVs can be broadly classified into different classes, namely, exosomes, microvesicles/microparticles, and apoptotic bodies [11]. They are thought to function primarily as intercellular communication vehicles to transfer bioactive cargoes to elicit diverse biological responses in recipient cells, and with MSC-EVs, many of these biological responses culminate in a therapeutic outcome in injured or diseased cells [12].

The current systematic review aims to summarize the results of the existing animal studies that were conducted to evaluate the therapeutic effects of MSCs and their secreted factors, including EVs, in animal models of TMJ cartilage/osteochondral defects and OA.

## 2. Materials and Methods

### 2.1. Purpose

The purpose of this study is to systematically review and evaluate the efficacy of MSCs, their secretome, and their secreted factors, including EVs, for TMJ repair in animal studies.

### 2.2. Systematic Review

This systematic review follows the recommendations of the Cochrane Handbook for Systematic Reviews of Intervention [13] and was conducted according to the Preferred Reporting Items for Systematic Reviews and Meta-Analysis (PRISMA) guidelines (Figure 1) [14,15].

### 2.3. Search Strategy

The literature search for eligible studies was conducted in PubMed, Embase, Scopus, and the Cochrane Library through 31 July 2023. The medical subject heading (MeSH) terms and keywords used were “temporomandibular joint” or “temporo-mandibular joint” or “temporo mandibular joint” or “craniomandibular” or “cranio-mandibular” or “cranio mandibular” or “TMJ” or “TMD” or “CMD” and “mesenchymal stem cell” or “mesenchymal stromal cell” or “marrow stem cell” or “marrow stromal cell” or “stromal cell” or “MSC” or “secretome” or “exosome” or “extracellular vesicle”.

### 2.4. Eligibility Criteria

The PICO framework was used to identify the components of evidence, which are as follows: Population: animals with TMJ condylar defects (cartilage, osteochondral) or TMJ-OA. Intervention: MSC-based therapies comprising MSCs, their secretome, or secreted factors, including EVs for TMJ repair. Comparison: healthy/naive controls, sham, and other groups (untreated, vehicle, scaffold). Outcome: the histological repair of TMJ and the associated structures as the primary outcome, and the morphological, molecular, and pain improvements as the secondary outcomes.

### 2.5. Study Selection

All search results were loaded into the EndNote 20 reference management software (Clarivate, Philadelphia, PA, USA), and duplicate records were removed. In the first stage, all abstracts and titles were retrieved and screened using the inclusion and exclusion criteria. After the initial screening, the full texts of the articles were further evaluated for eligibility. Three reviewers (Y.J., J.S., and W.D.) independently assessed all titles and abstracts for inclusion. Studies were included if they met the following criteria: in vivo animal studies that used MSCs, their secretome, or EVs for TMJ repair. Language was restricted to English. Studies were excluded if they were not related to TMJ or MSCs, not related to TMJ repair, or presented as case report, conference abstract, review, or purely in vitro study. As illustrated in Figure 1, a PRISMA flow diagram was used to document the study’s selection process [16].

### 2.6. Data Extraction

After identifying all the included studies, the following information was recorded: author; year of publication; animal species; age (weeks, months); sample size; gender (male or female); weight (grams or kilograms); animal model (cartilage/osteochondral defect, osteoarthritis); method of induction (surgically, mechanically, or chemically induced); source of MSCs, secretome, or EVs; isolation method; characterization, size distribution, and marker expression; experimental groups; study timepoints; concentration; volume; frequency; delivery; methods of analyses, including gross examination, imaging, histology, immunohistochemistry, molecular analysis, and pain measurement; as well as key in vitro and in vivo therapeutic outcomes. All qualitative and quantitative outcomes of the studies were recorded where available. Fate and biodistribution of MSCs, their secretome, or EVs, and their effects on cell proliferation, migration, and matrix synthesis, as well as immunomodulation, were also evaluated (Figure 2). Attempts were made to contact the corresponding author of the paper when certain details of the study were not reported.

### 2.7. Quality and Risk of Bias Assessments

The quality assessment for each included study was evaluated through their compliance with the ARRIVE (Animal Research: Reporting of In Vivo Experiments) guidelines [17]. Twenty items were evaluated for each study. The ratio for each item was calculated by dividing the category score (number of points assigned) by the total score (total points for the item). The quality of the item was assessed as “excellent” for values of the ratio between 0.8 and 1.0, “average” for values of the ratio between 0.5 and 0.79, and “low” for values of the ratio below 0.5. The risk of bias was evaluated using the Systematic Review Center for Laboratory Animal Experimentation (SYRCLE) risk of bias assessment tool for animal studies [18] that assessed selection bias, performance bias, detection bias, attrition bias, reporting bias, and other biases. The parameters assessed include sequence generation, baseline characteristics, allocation concealment, random housing, blinding of investigators, random outcome assessment, blinding of outcome assessor for the different analyses, incomplete outcome data, selective outcome reporting, and other sources of bias. All included studies were assessed as having high, unclear, or low risk of bias for the different parameters. This assessment was performed by first author (Y.J.) and independently checked by the second and third authors (J.S. and W.D.). Any discrepancy was resolved by discussion with the fourth author (K.Y.W.T.) and the senior author (W.S.T.).

## 3. Results

### 3.1. Search Process and Study Selection

The initial database search retrieved 2340 articles, which included 682 in PubMed, 1398 in Embase, 241 in Scopus, and 19 in Cochrane Library. A total of 749 duplicates were removed, and 1560 studies were excluded after screening the title and abstract. After reading the full texts of 31 articles that might meet the inclusion criteria, 6 articles on age-related degeneration, rheumatoid arthritis, osteoporosis, TMJ condylar resorption, growth, and replacement, as well as 2 articles of other languages, were excluded. A PRISMA flow chart showing the study-selection process is presented in Figure 1. A total of 23 studies that fulfilled the inclusion criteria were included in this systematic review [19,20,21,22,23,24,25,26,27,28,29,30,31,32,33,34,35,36,37,38,39,40,41]. They were published between 2011 and 2022 and more frequently (>50% of papers) between 2020 and 2022 (Figure 3). These 23 included studies were divided according to the nature of the assessed TMDs to enable reliable and equivalent comparisons between studies. These included 11 studies that evaluated MSC-based therapies for the repair of cartilage/osteochondral defects [19,20,21,22,23,24,25,26,27,28,29] and another 12 studies that evaluated MSC-based therapies for TMJ-OA treatment [30,31,32,33,34,35,36,37,38,39,40,41]. The outcomes of the studies were largely evaluated qualitatively as there were insufficient quantitative data across the studies for pooling of data quantitatively.

### 3.2. Basic Characteristics of the Included Studies

The details of all studies are described in Table 1, Table 2, Table 3, Table 4, Table 5 and Table 6. Table 1 details the animal species, age, sample size, gender, weight, animal model used, and method of induction. Table 2 details the source of MSCs, secretome, or EVs; isolation method; characterization; size distribution; and marker expression. Table 3 and Table 4 detail the comparison groups; study timepoints; concentration of MSCs, secretome, or EVs; volume; frequency; delivery; and methods of analysis, including gross examination, imaging, histology, immunohistochemistry, molecular analysis, and pain measurement. Table 5 and Table 6 summarize the key in vitro and in vivo outcomes reported in each paper.

### 3.3. Animal Models

To evaluate the efficacy of MSC-based therapies for the repair of cartilage/osteochondral defects, 2 main types of animal models were used. Namely, the condylar cartilage defect model was used in 1 study [20], and osteochondral defect models were used in 10 studies [19,21,22,23,24,25,26,27,28,29]. The defects were all surgically created, and the size of osteochondral defects ranged from a 3–5 mm diameter and 5 mm depth in *goats* [19,25] to a 2–3 mm diameter and 2–3 mm depth in *rabbits* [22,24,26,29] and 1 mm diameter in *rats* [23]. As for TMJ-OA, the efficacy of MSC-based therapies was most commonly evaluated in animal models that were chemically induced to develop OA via intra-articular injection of various agents such as bovine collagen II [32], mono-iodoacetate (MIA) [34,35,40], collagenase II [38,39], and combinations of Complete Freund’s adjuvant (CFA) and MIA [37] or interleukin (IL)-1β [41]. One study surgically induced OA via partial disc resection [30], whereas three studies mechanically induced TMJ-OA via unilateral anterior cross-bite (UAC) [31,33] or forced mouth opening [36]. Among the various animal species, *rabbits* were the most common animal species used in 10 studies [21,22,24,26,29,30,32,34,38,39] and accounted for 70.4% of animals used to create cartilage/osteochondral defects and 41.9% of animals used to induce experimental OA (Figure 4).

### 3.4. Source of MSCs, Their Derived Secretome, or EVs

A total of 10 studies used *human* MSCs or their derivatives [21,23,28,29,34,35,36,37,38,39], 3 studies used *goat* MSCs [19,20,25], 5 studies derived MSCs from *rabbits* [22,24,26,30,32], 2 studies used *rat* MSCs or their derivatives [40,41], and three studies derived MSCs from *mice* [27,31,33]. Among the various MSC sources that included bone marrow, adipose tissue, dental pulp, exfoliated deciduous teeth, umbilical cord, and embryonic stem cells, bone marrow stem/stromal cells (BMSCs) were the most commonly used cell source for the treatment of TMJ cartilage/osteochondral defects [19,20,22,24,25,26,28] and OA [30,31,32,33,38,40,41]. In terms of the use of cells, secretome, or EVs, eighteen studies tested MSCs as cell-based therapies [19,20,21,22,23,24,25,26,27,28,30,31,32,33,34,37,39,41], of which two studies reported the co-culture of MSCs and chondrocytes [20,26]. The other five studies investigated the use of MSC-derived secretome [36] and EVs [29,35,38,40]. Regardless of the origin or source of MSCs, all studies reported the therapeutic efficacy of MSCs, their derived secretome, or EVs in treating TMJ cartilage/osteochondral defects [19,20,21,22,23,24,25,26,27,28,29] and OA [30,31,32,33,34,35,36,37,38,39,40,41] in vivo. In comparing the different sources of MSCs, Kim et al. [34] compared *human* umbilical cord MSCs (UCMSCs) and BMSCs in their proliferation and chondrogenic differentiation capabilities in vitro. Of note, *human* UCMSCs had a comparable proliferation capacity as the *human* BMSCs and demonstrated a marked chondrogenic differentiation capacity, characterized by the presence of lacunae containing chondrocytes embedded in the matrix. Additionally, MSCs may also be co-cultured with additional cell sources such as chondrocytes [20,26]. By co-culturing with chondrocytes, a chondrogenic microenvironment was fostered to induce chondrogenic differentiation of MSCs without the additional factors [20].

### 3.5. Isolation and Characterization of MSCs, Their Derived Secretome, or EVs

Most of the studies used centrifugation with or without density gradients for the isolation of mononuclear cells, followed by the separation of MSCs via plastic adherence to tissue culture flasks [19,20,22,24,25,26,28,30,32,34,37,41]. One study, however, used a commercially available isolation system (Celution 800/CRS system) to isolate *human* adipose-derived regenerative cells (ADRCs) with varying proportions of stem cells, smooth muscle cells, endothelial cells, hematopoietic cells, nucleated cells, and other stromal cells from the *human* adipose tissue [21]. For the characterization of MSCs, flow cytometry and/or multi-lineage differentiation assays coupled with staining and gene expression analysis were frequently performed to analyze MSC marker expression and differentiation to osteogenic, chondrogenic, and/or adipogenic lineages [22,23,25,26,27,28,29,32,34,39]. Despite some variability among studies, the MSCs were generally positive for CD29, CD44, CD73, CD90, and CD105 and negative for CD31 and CD45 [22,23,28,29,32,34,39,40]. Unlike MSCs, secretome was characterized by liquid chromatography–tandem mass spectrometry (LC-MS/MS) analysis, whereas EVs were typically characterized by nanoparticle tracking analysis (NTA), transmission electron microscope (TEM), and/or Western blotting in the analysis of their size distribution, morphology, and presence of EV markers [29,35,38,40]. These EVs generally appeared as bi-lipid membrane structures under TEM, displayed a size distribution range of approximately 79–200 nm, and expressed endosomal proteins, ALIX and TSG101, and tetraspanin proteins, CD9, CD63, and CD81 [29,35,38,40].

### 3.6. Concentration

A wide range of concentrations of MSCs and their derived secretome or EVs were employed across different studies. The concentration of MSCs ranged from 10^5^ cells/mL [28,31,33] to 7.5 × 10^7^ cells/mL [19,20,24,25,26,27,28,31,33,34,37,39]. Of these, one study evaluated the therapeutic efficacy of *human* UCSCs in a *rabbit* model of TMJ-OA, comparing 5 × 10^5^, 2.5 × 10^6^, and 5 × 10^6^ cells/mL [34]. All three cell concentrations at single injections showed anti-inflammatory effects and cartilage regenerative capacity, although the medium concentration of MSCs (2.5 × 10^6^ cells/mL) was most effective in regenerating the articular cartilage and showed the highest gene expression levels of growth factors, including insulin growth factor (IGF)-1 and fibroblast growth factor (FGF)-2 [34].

EVs are commonly measured by their particle and/or protein concentration. Among the studies that investigated the use of EVs, one study reported particle concentration of EVs around 2–4 × 10^9^ particles/mL [38], two studies reported protein concentration of about 2 mg/mL [35,40], and one study loaded 5 µg EVs per scaffold for implantation [29]. With the use of secretome, Ogasawara et al. [36] adjusted the protein concentration of the conditioned medium to 3 μg/mL with serum-free Dulbecco’s Modified Eagle Medium (DMEM).

### 3.7. Delivery

A variety of biomaterial scaffolds/carriers, ranging from platelet derivatives [22,23,28], collagen, gelatin, and hyaluronic acid (HA)-based hydrogels [24,26,27] to silk fibroin [29], poly(lactic-co-glycolic acid) (PLGA), and polycaprolactone (PCL)-based polymeric scaffolds [19,25] have been used to deliver MSCs or their EVs for the repair of cartilage/osteochondral defects. Biphasic scaffolds were also designed to simulate the specific structures and matrix compositions of the osteochondral tissues [25,26]. Of note, Wang and colleagues designed the biomimetic biphasic scaffold comprising self-crosslinking thiolated hyaluronic acid (HA-SH) and collagen I blend hydrogel and biphasic calcium phosphate (BCP) ceramics [26]. The blend hydrogel was layered and embedded with chondrocytes and BMSCs to simulate stratifications of the cartilage tissue and BCP ceramics to facilitate bone formation [26]. Interestingly, Putnova et al. flushed the defect with the *human* ADRC solution without using any scaffold [21].

For the treatment of TMJ-OA, most of the studies delivered MSCs, their derived secretome, or EVs by injections using various carriers, ranging from DMEM/10% fetal calf serum (FCS) [31,33] and serum-free DMEM [36] to phosphate-buffered saline (PBS)/saline [32,34,35,37,38,40], and hylartin/HA [30,39]. One study, however, used the gelatin methacryloyl (GelMA) hydrogel microspheres loaded with transforming growth factor (TGF)-β for coating with *rat* BMSCs [41]. In that study, the sustained release of TGF-β induced chondrogenic differentiation of BMSCs for enhanced TMJ repair in a *rat* model of TMJ-OA [41].

All the studies that evaluated scaffold/carrier delivery reported superior therapeutic outcomes in TMJ repair with scaffold/carrier delivery of MSCs, their derived secretome, or EVs against scaffolds alone [19,20,22,23,24,25,26,29,30,39,41]. Regarding the frequency of treatment, most of the studies applied a single scaffold implantation with MSCs [19,20,22,23,24,25,26,27,28] or MSC-EVs [29] or a single injection of MSCs [30,32,34,37,39,41] or MSC-EVs [38,40]. Other studies applied multiple injections [31,33,35,36]. None of the studies compared the efficacy of different biomaterial scaffolds or varying treatment frequencies for cartilage regeneration in their animal models. The optimal delivery method and frequency of treatment remain to be determined, and further studies for optimization might be needed.

### 3.8. Morphological Outcomes

Morphological analyses were mainly performed via gross/macroscopic assessment, micro-computed tomography (micro-CT), and X-ray examination. Eight studies performed gross examination and reported improved defect filling, macroscopic appearance, and tissue integration in cartilage/osteochondral defects [19,20,21,22,24,25,26,29], whereas one study noted improved integrity of the osteoarthritic condyles [38] following treatment with MSCs [19,20,21,22,24,25,26] or their EVs [29,38]. Micro-CT or X-ray examinations further observed improved subchondral bone reconstitution and restoration in cartilage/osteochondral defects [19,20,24,26,29] and in OA [30,31,34,35,36,37,38,41]. In these studies, a significantly higher bone mineral density (BMD), higher percentage of bone volume over total volume (BV/TV), increased trabecular thickness (Tb.Th), increased trabecular bone number (Tb.N), decreased trabecular separation (Tb.Sp), and/or reduced bone surface over bone volume ratio (BS/BV) were observed in animals treated with MSCs [19,20,24,26,30,31,37,41], secretome [36], or EVs [29,35,38].

### 3.9. Histological Outcomes

All studies performed histological analysis where specimens were stained with hematoxylin and eosin [20,21,22,23,25,26,27,28,29,31,32,33,34,35,36,37,38,40,41] for the observation of the general morphology; with safranin O [19,24,25,26,27,29,30,31,32,33,34,38,39], alcian blue [21,28,38], and toluidine blue [22,30,35,36,37] for glycosaminoglycan (GAG); and with Masson [26] or picrosirius red [39] for collagen deposition. Tartrate-resistant acid phosphatase (TRAP) [31,36] staining was also performed to detect bone-absorbing osteoclasts and terminal deoxynucleotidyl transferase (TdT) dUTP nick-end labeling (TUNEL) [36] for apoptotic cells. Immunohistochemistry was frequently performed to detect cartilage matrix proteins, namely, collagen I, collagen II, and aggrecan [19,20,23,24,25,26,29,30,31,33,34,38]; proliferative cell nuclear antigen (PCNA) or antigen Kiel 67 (Ki-67) for proliferative cells [23,33,35,36,38]; and matrix metalloproteinases (MMPs) for matrix degradation [35,36,37,38] and pro-inflammatory cytokines, including interleukin (IL)-1β, interferon (IFN)-γ, and/or tumor necrosis factor (TNF)-α for inflammation [35,36,37].

In the TMJ cartilage/osteochondral defect models, animals treated with MSCs or their EVs generally showed improved defect filling with cartilaginous tissue, smooth surface regularity, and integration with adjacent condylar cartilage, as well as subchondral bone reconstitution [19,20,22,23,24,25,26,27,28,29]. Of these studies, six studies conducted histological scorings and reported significantly improved scores in groups treated with MSCs compared with untreated and/or scaffold groups [19,20,22,24,26,28].

In the TMJ-OA models, animals treated with MSCs, their derived secretome, or EVs had reduced degenerative changes and showed enhanced cartilage and subchondral bone restoration with increased matrix deposition and improved structural integrity [30,31,32,34,35,36,37,38,39,40,41]. These therapeutic effects were often associated with increased levels of PCNA for proliferation [35,36,38]; decreased levels of IL-1β, TNF-α, IFN-γ for inflammation [35,36,37]; MMP3 and MMP13 for matrix degradation [35,36,37,38]; TRAP for subchondral bone resorption [31,36]; as well as reduced levels of TUNEL, cleaved caspase (CCP)3, and death-associated protein (DAP)3 for apoptosis and anoikis [33,35,36]. Of these studies, six studies performed histological scorings to assess synovial inflammation and cartilage and subchondral bone destruction. They reported significantly improved scores in animals treated with MSCs, their derived secretome, or EVs, with reduced synovial inflammation, improved cartilage cellularity, and matrix deposition compared with the vehicle-treated and/or untreated animals [30,34,35,36,37,38].

### 3.10. Molecular Outcomes

Six studies performed gene expression analysis of the condylar cartilage specimens [22,30,31,34,35,40]. Using a *rabbit* TMJ osteochondral defect model, Cheng et al. reported significantly upregulated expression of cartilage-specific genes, including *Sox9*, *Col II*, and *ACAN*, in the repaired cartilage following treatment with a cell sheet of BMSCs, and this upregulation was further enhanced when the cells were pre-treated with hydrostatic pressure and delivered in a platelet-rich fibrin (PRF) scaffold [22]. In the TMJ-OA models, five studies reported a significantly upregulated expression of cartilage-specific genes (*Sox9*, *Col2a1*, and *ACAN*) and downregulated expression of genes associated with inflammation (*IL-1β*, *IL-17*, *TNF-α*, and *NF-κB*) and matrix degradation (*MMP3*, *MMP13*, and *ADAMTS5*) in animals treated with MSCs or their derived EVs compared with the vehicle-treated and/or untreated animals [30,31,34,35,40]. Notably, there was also increased expression of genes such as *TGF-β1*, *IGF-1*, and *FGF-2*, which are growth factors involved in chondrogenic differentiation and cartilage anabolism [34]. One study also reported downregulated levels of genes associated with fibrosis (*α-SMA*), apoptosis (*BAX*), and pain (*Substance P*, *CGRP*, *NGF*, *p75NTR*, and *TrkA*) with EV treatment compared to those of vehicle treatment [35].

### 3.11. Pain Behavioral Outcomes

In three studies, the head withdrawal threshold was measured to evaluate the nociceptive responses of the animals during the course of treatment. In these studies, regardless of the route of administration (intra-articular vs. intravenous), MSCs and their derived secretome, including EVs, were all found to be effective in relieving hyperalgesia of progressive TMJ-OA in *mice* and *rats* [35,36,37]. However, it is important to note that the secretome and EVs were applied via multiple injections, whereas the cells (i.e., DPSCs) were applied via a single intra-articular injection [35,36,37].

### 3.12. Fate and Biodistribution of MSCs, Their Derived Secretome, or EVs

Among the 18 studies that conducted implantation or intra-articular injections of MSCs, 6 studies performed in vivo tracing of MSCs [19,20,24,31,33,34]. In cartilage/osteochondral defects, transplanted MSCs were found in the newly formed tissues for up to 6 weeks [19] and 12 weeks post-surgery [20] in *goats* and up to 12 weeks in *rabbits* [24]. On the other hand, two studies reported a rapid decline in transplanted MSCs after 7 days and complete loss of the transplanted cells by 20 days following transplantation in a UAC-induced OA *mouse* model [31,33]. One study, however, reported the presence of transplanted MSCs up to 4 weeks post-injection in a *rabbit* TMJ-OA model induced by MIA [34]. Based on these observations, the persistence of MSCs following administration may vary in cartilage/osteochondral defects and in OA in the different animal models. In contrast to the MSC studies, none of the secretome or EV studies performed in vivo tracing of the secretome factors or EVs [29,35,36,38,40].

### 3.13. Cellular Proliferation, Migration, and Matrix Synthesis

Five studies reported increased proliferation and/or decreased apoptosis or anoikis following treatment with MSCs, their derived secretome, or EVs [23,33,35,36,38]. One study reported the upregulated expression of stromal-derived factor (SDF)-1 and regulated upon activation normal T-cell expression and secreted RANTES via osteoarthritic cartilage, and thus a higher capacity of attracting the migration of MSCs toward the degraded cartilage in OA [31]. These in vivo findings were supported by the in vitro studies where EV treatment was found to enhance the proliferation and migratory activity of BMSCs and chondrocytes in vitro [29,38]. Notably, EVs derived from inflammation-stimulated MSCs significantly enhanced the migration of BMSCs over their unstimulated counterpart [29].

Regardless of cartilage/osteochondral defects or OA, most of the studies reported increased matrix synthesis and deposition following treatment with MSCs, their derived secretome, or EVs [19,20,22,23,24,25,26,27,28,29,30,31,32,33,34,35,36,37,38,39]. In TMJ-OA studies, concurrent attenuation of MMP3 [37,40], MMP13 [30,31,35,36,37,38,40], and a disintegrin and metalloproteinase with thrombospondin motifs 5 (ADAMTS5) [35] were also observed. In support of the in vivo findings, three studies also reported significant inhibition of MMP3 and/or MMP13 expression in condylar chondrocytes and synoviocytes under inflammation following treatment with MSCs or their EVs [35,37,38]. These studies further identified protein kinase B (AKT), extracellular signal-regulated kinase (ERK), adenosine monophosphate-activated protein kinase (AMPK), signal transducer and activator of transcription 1 (STAT1), and Hippo-YAP (Yes-associated protein) signaling pathways in regulating the anabolic activities of MSCs and their EVs [35,37,38]. For instance, co-culture with DPSCs inhibited the expression of MMP3 and MMP13 in synoviocytes under inflammation, but these effects were abolished by the STAT1 inhibitor, implicating the role of the STAT1 signaling pathway in regulating the expression of MMP3 and MMP13 [37]. In another study, MSC-EVs were found to enhance GAG synthesis and inhibited nitric oxide and MMP13 production in IL-1β-treated chondrocytes through adenosine receptor activation of AKT, ERK, and AMPK phosphorylation [35].

### 3.14. Immunomodulation

Two studies examined the immune cell infiltration during TMJ repair [21,37]. Reduced infiltration of CD4+ T cells was observed in the synovial tissue and subchondral bone marrow of OA *rats* treated with *human* DPSCs compared to those treated with saline [37]. On the other hand, Putnova et al. observed significant immune cell infiltration and inflammatory reaction in both soft and hard TMJ tissues in *rabbits* treated with *human* ADRCs despite prior immunosuppression, and these were attributed to the heterogeneous cell populations present in ADRCs [21]. In other studies, pro-inflammatory mediators, including IL-1β, TNF-α, IFN-γ, IL-17, and inducible nitric oxide synthase (iNOS) [31,34,35,36,37,40] were downregulated, while anti-inflammatory cytokines such as TGF-β1 and IL-10 [34] were upregulated in animals treated with MSCs, their derived secretome, or EVs compared with the vehicle-treated and/or untreated animals. In vitro, EVs derived from inflammation-stimulated MSCs demonstrated augmented efficacy in suppressing inflammation and repolarizing the THP-1 cells toward the M2-like phenotype with the expression of M2 markers, including CD206, IL-10, and CCL22 [29]. These effects of MSC-EVs on the macrophage phenotype and polarization were attributed to the high expression level of miR-27b-3p that regulated macrophage polarization by targeting macrophage colony-stimulating factor (CSF)-1 [29].

### 3.15. Compliance with the ARRIVE Guidelines

The compliance with the ARRIVE guidelines for all the studies was evaluated and is detailed in Table 7 and Table 8. Based on the category score/total score ratio, three items (1, 2, and 20) were assessed as excellent, eleven items (3, 4, 5, 6, 7, 8, 10, 11, 13, 16, and 19) were marked as average, and the remaining six items (9, 12, 14, 15, 17, and 18) were considered low for studies on TMJ cartilage/osteochondral defects. Consequently, the overall quality score was 0.586. As for the studies on TMJ-OA, five items (1, 5, 6, 13, and 20) were assessed as excellent, nine items (2, 3, 7, 8, 10, 11, 14, 16, and 19) were marked average, and the remaining items (4, 9, 12, 15, 17, and 18) were considered low. Consequently, the overall quality score for the TMJ-OA studies was 0.609, considered average quality. Among all, five items (9, 12, 15, 17, and 18) were consistently marked low in both studies for cartilage/osteochondral defects and TMJ-OA.

### 3.16. Risk of Bias Assessment

The SYRCLE risk of bias tool was used to assess the risk of bias for all the included studies, and the assessment is summarized in Table 9. Most domains had unclear risks due to the lack of information (Table 9). None of the studies reported random sequence generation under selection bias and random housing and blinding under performance bias, and therefore were assigned unclear risk [19,20,21,22,23,24,25,26,27,28,29,30,31,32,33,34,35,36,37,38,39,40,41]. Most studies reported some form of allocation concealment and baseline characteristics, including the species, strain, sex, or weight of the animals used, and were therefore assigned low risk. Only three studies were assigned a low risk for detection bias secondary to the random selection of animals for outcome assessment [22,24,30], while the rest of the studies were classified as unclear risk [19,20,21,23,25,26,27,28,29,31,32,33,34,35,36,37,38,39,40,41]. Only one study reported blinding of the assessor when performing gross assessment [29], whereas nine studies stated that the assessor was blinded when performing the histological analysis [19,22,29,30,34,35,37,38,39], and therefore were classified as low risk under detection bias. However, those studies that performed imaging, immunohistochemical, and/or molecular analyses did not report blinding for detection bias and therefore presented unclear risk. Notably, all studies were free of selective outcome reporting and presented no apparent issues that could result in a high risk of bias [19,20,21,22,23,24,25,26,27,28,29,30,31,32,33,34,35,36,37,38,39,40,41].

## 4. Discussion

The principal finding of this systematic review is that MSC-based therapies are efficacious in the treatment of TMJ cartilage/osteochondral defects and OA. This therapeutic efficacy of MSC-based therapies in TMJ repair is demonstrated by cartilage regeneration and subchondral bone restoration, with overall improvements in the morphological, histological, molecular, and behavioral pain outcomes in the studies reviewed [19,20,21,22,23,24,25,26,27,28,29,30,31,32,33,34,35,36,37,38,39,40,41].

Of 23 studies reviewed, 18 studies tested MSCs as cell-based therapies [19,20,21,22,23,24,25,26,27,28,30,31,32,33,34,37,39,41], and the other five studies investigated the use of MSC-derived secretome [36] and EVs [29,35,38,40] as “cell-free” therapies. Of note, these five studies were published in recent years from 2019 to 2022 and signal a paradigm shift in the therapeutic mechanism of MSCs in tissue repair from one based on cellular differentiation and replacement to one based on secretion and paracrine signaling. Indeed, the role of paracrine secretion, particularly EVs, in mediating the wide-ranging therapeutic efficacy of MSCs has been increasingly reported [42], and it is now accepted that MSCs exert many if not most of their paracrine effects through the release of EVs [43], membrane vesicles that are secreted by all cell types.

Despite the therapeutic efficacy of MSC-based therapies, several factors/variables have been identified to influence the outcome of intervention with the use of MSCs, their secretome, or EVs for TMJ repair. These factors primarily include the cell source, dosage/concentration, and delivery. Several MSC sources were demonstrated to be efficacious in TMJ repair in both cartilage/osteochondral defects and OA, but the ideal cell source is yet to be identified. Kim et al. demonstrated comparable proliferation and differentiation capabilities of *human* UCMSCs and BMSCs in vitro; however, this remains to be verified in an animal model in vivo [34]. A head-to-head comparison of relevant sources of MSCs, their derived secretome and/or EVs would be required in future studies to determine the most effective source from a TMJ repair perspective. Similarly, a wide range of concentrations of MSCs and their derived EVs was reported. One study evaluated a single intra-articular injection of *human* UCSCs at varying concentrations in a *rabbit* model of TMJ-OA and identified the medium concentration of MSCs (2.5 × 10^6^ cells/mL) as the optimal dose/concentration for regenerating the articular cartilage [34]. However, this dose/concentration remains to be validated using other sources of MSCs.

The frequency of treatment is another factor/variable that should be considered and ideally tested. While some studies recorded the long-term survival of the transplanted BMSCs up to 12 weeks in the osteochondral defects of *rabbit* and *goat* [20,24], other studies reported a rapid decline in transplanted cells after 1 week in OA in *mice* and required multiple weekly injections to constantly supply the BMSCs [31,33]. Based on these observations, the persistence of MSCs following implantation or intra-articular injection may vary in cartilage/osteochondral defects and in OA in different animal models. In contrast, none of the secretome or EV studies performed in vivo tracing of the secretome factors or EVs. As such, it would be necessary to determine in future studies the fate and bio-distribution of MSCs, their secretome, and EVs following injection into the joint space to determine their site of action and persistence and to optimize the number of injections for optimal treatment.

Other factors, such as the culture condition and the use of a scaffold, would also influence the treatment outcome in TMJ repair. The culture condition affects the potency of MSCs, their derived secretome, and EVs. For instance, in vitro chondrogenic pre-differentiation or hydrostatic pressure stimulation substantially enhanced the chondrogenic potential of MSCs and, henceforth, their therapeutic efficacy in TMJ repair [22,30]. The overexpression of NEL-like protein-1 (NELL-1) or hypoxia-inducible factor (HIF)-1α also reportedly enhanced the efficacy of MSCs in osteochondral repair. In those studies, animals treated with NELL-1 or HIF-1α overexpressing BMSCs displayed enhanced fibrocartilage regeneration and subchondral bone restoration with increased matrix deposition and improved structural organization compared to those treated with native MSCs [19,24]. Notably, HIF-1α overexpression was found to enhance the survival of transplanted BMSCs, contributing to improved cartilage repair [24]. Alternatively, EVs derived from MSCs stimulated under inflammation in the presence of TNF-α and IFN-γ demonstrated augmented efficacy for osteochondral repair compared to those derived from unstimulated counterparts [29]. Other studies also reported the delivery of MSCs and their derived EVs in different scaffolds/carriers and demonstrated synergistic enhancements in cartilage and subchondral bone repair [19,20,22,23,24,25,26,27,28,29,30,39,41]. For example, Cheng et al. fabricated a BMSC sheet in the PRF scaffold and showed that the BMSC/PRF construct outperformed the BMSC sheet or PRF alone in cartilage regeneration by enhancing matrix synthesis and promoting the mechanical properties and integration of the neocartilage [22]. These studies offer possible strategies to engineer and/or coax the MSCs and their derived EVs to possess the desired therapeutic properties for enhanced TMJ repair. However, it is also important to note that these findings of a particular modification or scaffold being superior to another are all derived from single studies [19,20,22,23,24,25,26,27,28,29,30,39,41], and the conclusions remain to be verified by more studies.

Of the 23 studies reviewed, nine studies applied *human* MSCs, their derived secretome, or EVs in immunocompetent animals, and no adverse responses were reported [23,28,29,34,35,36,37,38,39]. These findings are consistent with the well-established immune privilege property of MSCs. The cells and their derived secretome, including EVs, did not elicit any adverse responses from the animals, including *mice*, *rats*, and *rabbits* tested in these studies [23,28,29,34,35,36,37,38,39]. These findings suggest the use of MSCs, their derived secretome, and EVs as potential allogeneic therapies.

Although the therapeutic mechanisms underlying MSC-based therapies in TMJ repair remain to be elucidated, it could be summarized from this systematic review of 23 studies that MSCs, including their secretome and EVs, likely exert a multifaceted activity in TMJ repair by enhancing cellular proliferation, migration, and matrix synthesis and modulating the immune reactivity (Figure 5). However, in the 23 studies reviewed, only 3 studies delved into the underlying signaling pathways [35,37,38], and only 2 studies investigated immune cell infiltration during TMJ repair [21,37]. These clearly reflect our nascent understanding of the therapeutic mechanisms, particularly the immunomodulatory mechanisms, underlying the effects of MSCs, their derived secretome, or EVs in TMJ repair, which warrants further investigation. The identification of key factors/variables influencing the therapeutic outcomes of MSCs, their secretome, and EVs in TMJ repair and improved understanding of the underlying mechanisms, as summarized in this timely review, will undoubtedly inform the planning and design of future studies.

This systematic review has several limitations. Although most of the studies included in this review appeared to be methodologically sound, the lack of proper reporting rendered many of these studies as having unclear risk in several domains, such as random sequence generation; random housing during the experiment; the blinding of caregivers and investigators; and the blinding of outcome assessors in gross, imaging, and immunohistochemical and molecular analyses, according to the SYRCLE risk of bias assessment tool. This highlighted the importance of detailed reporting by adherence to the ARRIVE guidelines so as to improve the credibility and reliability of studies as having high-quality evidence. Several factors/variables, including the cell source, dosage/concentration, scaffold, and delivery route, were also identified to influence the therapeutic outcome of MSCs, their secretome, or EVs in TMJ repair. However, none of these studies systematically evaluated these variables to determine the optimal cell source, concentration, scaffold, delivery route, and/or frequency of treatment, which would likely require optimization for a specific TMJ cartilage/osteochondral defect or OA condition. The deficiency in quantitative data and heterogeneity in the assessment and reporting of outcomes in many studies also precluded a more rigorous meta-analysis of the studies in this review. Therefore, more efforts to standardize the reporting of methodology and outcomes are required in future studies. This would then allow future studies to have better uniformity and validity, with the possible pooling of results through meta-analysis for more robust data analysis and conclusion.

## 5. Conclusions

In this review, we systematically assessed the existing preclinical animal studies and broadly demonstrated the effectiveness of MSC-based therapies for TMJ repair in animal models of TMJ cartilage/osteochondral defects and OA. In general, MSC-based therapies were found to exert positive effects on cell proliferation, migration, matrix synthesis, and immunomodulation, leading to improvements in morphological, histological, molecular, and behavioral pain outcomes.

## Figures and Tables

**Figure 1 cells-13-00990-f001:**
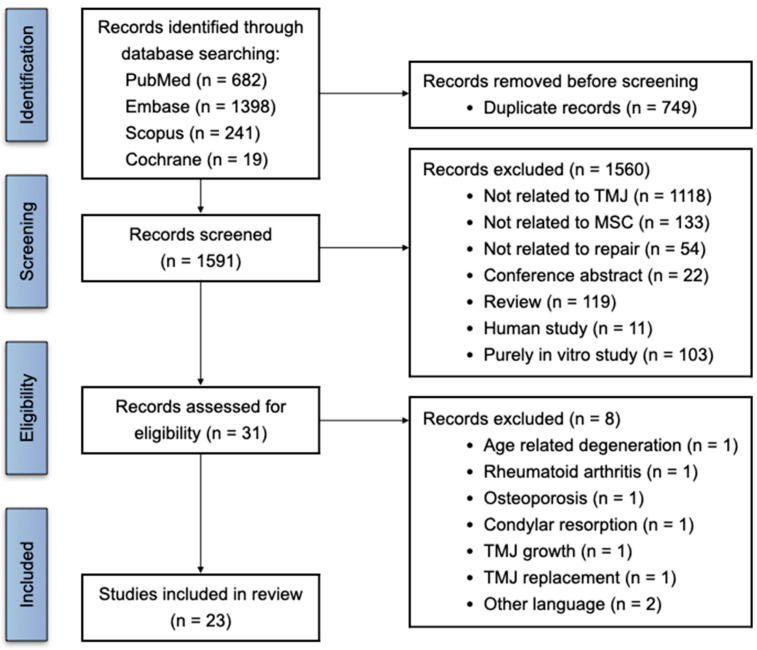
Prisma flow diagram. Search strategy and selection process of the included studies.

**Figure 2 cells-13-00990-f002:**
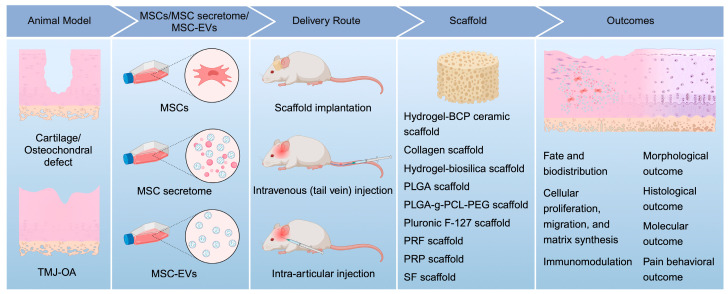
Key factors and outcomes identified in preclinical studies of MSC-based therapies for TMJ repair. BCP, biphasic calcium phosphate; EVs, extracellular vesicles; MSCs, mesenchymal stem/stromal cells; PCL, polycaprolactone; PEG, polyethylene glycol; PLGA, poly(L-glutamic acid); PRF, platelet-rich fibrin; PRP, platelet-rich plasma; SF, silk fibroin; TMJ-OA, temporomandibular joint osteoarthritis.

**Figure 3 cells-13-00990-f003:**
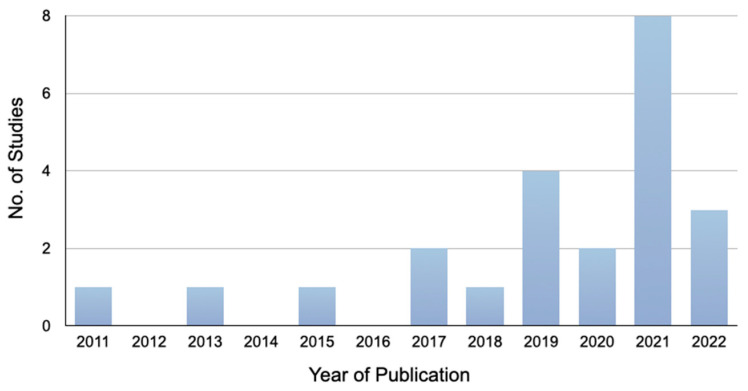
Bar chart showing the number of published studies included in the systematic review sorted by the year of publication.

**Figure 4 cells-13-00990-f004:**
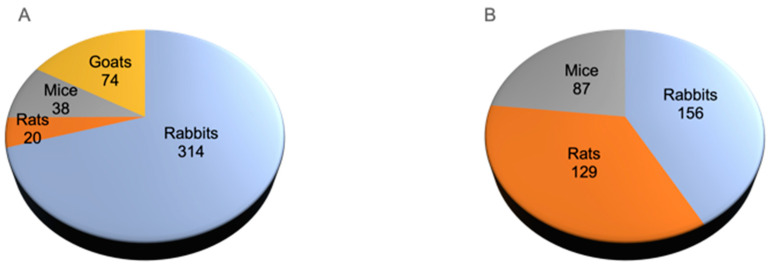
Animal species used for models of (**A**) TMJ cartilage/osteochondral defects and (**B**) OA.

**Figure 5 cells-13-00990-f005:**
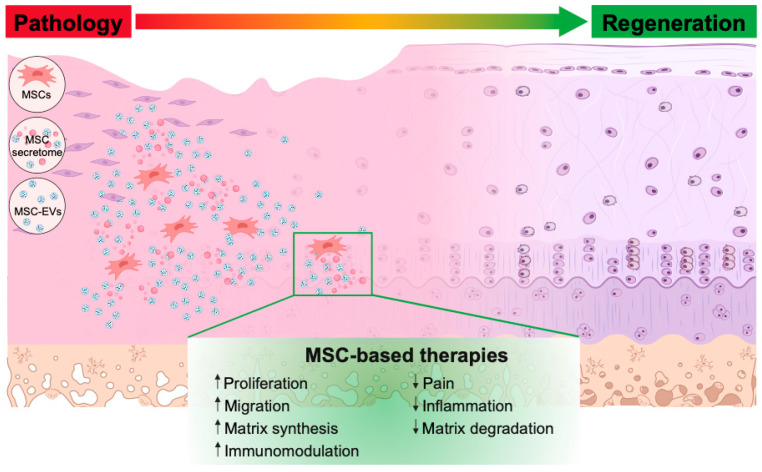
MSC-based therapies alleviate pathological processes in TMJ cartilage/osteochondral defects and OA through a multifaceted mechanism of enhancing cellular proliferation, migration, and matrix synthesis and modulating the immune reactivity.

**Table 1 cells-13-00990-t001:** Summary of characteristics of animal models.

Author, Year	Animal	Age	Sample Size	Gender	Weight	Animal Model	Method of Induction
Zhu, 2011 [19]	Eastern cross *goats*	25–28 months	50	Male	32–37 kg	Osteochondral defect	Surgical; 3 mm diameter, 5 mm depth
Sun, 2018 [20]	*Goats*	6–8 months	12	Male	10–22 kg	Condylar cartilage defect	Surgical; full-thickness condylar cartilage defect
Putnova, 2019 [21]	NZ *rabbits*	10 months	17	Male	NR	Osteochondral defect	Surgical; NR
Cheng, 2019 [22]	NZ *rabbits*	6 months	180	Male	NR	Osteochondral defect	Surgical; 3 mm diameter, 3 mm depth
Sumarta, 2021 [23]	Wistar *rats*	3 months	20	Male	200–300 g	Osteochondral defect	Surgical; 1 mm diameter
Cheng, 2021 [24]	NZ *rabbits*	4 months	45	Male	2–3 kg	Osteochondral defect	Surgical; 3 mm diameter, 2 mm depth
Yu, 2021 [25]	*Goats*	12 months	12	NR	20 kg	Osteochondral defect	Surgical; 5 mm diameter, 5 mm depth
Wang, 2021 [26]	NZ *rabbits*	7 months	16	Male	3 kg	Osteochondral defect	Surgical; 2 mm diameter, 3 mm depth
Guastaldi, 2021 [27]	C57BL/6 *mice*	8–10 weeks	30	Female	NR	Osteochondral defect	Surgical; linear condylar cartilage defect
Gomez, 2021 [28]	C57BL/6 *mice*	8–10 weeks	8	Female	NR	Osteochondral defect	Surgical; focal cartilage defect
Liu, 2022 [29]	NZ *rabbits*	NR	56	NR	2.2–2.5 kg	Osteochondral defect	Surgical; 2 mm diameter, 2 mm depth
Chen, 2013 [30]	NZ *rabbits*	6 months	46	NR	2.5–3.2 kg	TMJ-OA	Surgical; disc resection
Lu, 2015 [31]	C57BL/6J *mice*	6 weeks	27	Female	17–19 g	TMJ-OA	Mechanical; UAC
Zaki, 2017 [32]	NZ rabbits	NR	50	Male	1–1.5 kg	TMJ-OA	Chemical; bovine collagen II injection
Zhang, 2017 [33]	C57BL/6J *mice*	6 weeks	40	Female	17–19 g	TMJ-OA	Mechanical; UAC
Kim, 2019 [34]	NZ *rabbits*	NR	25	Male	2.5–2.8 kg	TMJ-OA	Chemical; MIA injection
Zhang, 2019 [35]	SD *rats*	8 weeks	48	Female	198–271 g	TMJ-OA	Chemical; MIA injection
Ogasawara, 2020 [36]	*Mice*	11 weeks	20	Male	NR	TMJ-OA	Mechanical; forced mouth opening
Cui, 2020 [37]	SD *rats*	7 weeks	15	Female	180–200 g	TMJ-OA	Chemical; CFA and MIA injection
Wang, 2021 [38]	NZ *rabbits*	12–18 weeks	7	Male and female	NR	TMJ-OA	Chemical; Collagenase II injection
Köhnke, 2021 [39]	*Rabbits*	12 weeks	28	Female	800 g	TMJ-OA	Chemical; Collagenase II injection
AbuBakr, 2022 [40]	Albino *rats*	3–4 months	48	Male	150–170 g	TMJ-OA	Chemical; MIA injection
Yang, 2022 [41]	SD *rats*	6 weeks	18	Male	NR	TMJ-OA	Chemical; CFA combined with IL-1β injection

CFA, Complete Freund’s adjuvant; IL-1β, interleukin-1β; MIA, monosodium iodoacetate; NZ, New Zealand; NR, not reported; SD, Sprague Dawley; TMJ-OA, temporomandibular joint osteoarthritis; UAC, unilateral anterior cross-bite.

**Table 2 cells-13-00990-t002:** Characterization of MSCs and their derived secretome and EVs in vitro.

Author, Year	Source	Isolation Method	Characterization	Size Distribution	Marker Expression
Zhu, 2011 [19]	*Goat* BMSCs	Density gradient centrifugation, adherence to tissue culture flask	NR	NA	ND
Sun, 2018 [20]	*Goat* BMSCs and auricular chondrocytes	Centrifugation, adherence to tissue culture flask	Phase contrast microscopy	NA	ND
Putnova, 2019 [21]	*Human* ADRCs	Celution 800/CRS system	NR	NA	CD146+, CD34+, CD31+, CD45+
Cheng, 2019 [22]	*Rabbit* BMSCs	Centrifugation, adherence to tissue culture flask	Flow cytometry, adipogenic and osteogenic differentiation, ARS, ALP, oil red O staining, TEM, SEM	NA	CD34−, CD45−, CD29+ and CD44+
Sumarta, 2021 [23]	*Human* UCMSCs	NR	Immunocytochemical staining, flow cytometry	NA	CD45−, CD73+, CD90+, CD105+
Cheng, 2021 [24]	*Rabbit* BMSCs	Centrifugation, adherence to tissue culture flask	Immunofluorescence, WB analysis for HIF-1α expression	NA	HIF-1α
Yu, 2021 [25]	*Goat* BMSCs	Centrifugation, adherence to tissue culture flask	Chondrogenic induction, CLSM, SEM, gene expression analysis	NA	Col I, Col II, Sox9
Wang, 2021 [26]	*Rabbit* BMSCs and articular chondrocytes	Centrifugation, adherence to tissue culture flask	Live/dead, osteogenic and chondrogenic differentiation, ALP, ARS, SEM, GAG/DNA, gene expression analysis	NA	ACAN, Sox9, Col1a2, Col2a1, Col10a1
Guastaldi, 2021 [27]	*Mouse* TMJ condyle-derived MSCs	NR	Live/dead, flow cytometry	NA	NR
Gomez, 2021 [28]	*Human* BMSCs	Ficoll-Hypaque isolation, adherence to tissue culture flask	Flow cytometry, osteogenic and chondrogenic differentiation, ARS, AB staining	NA	CD90+, CD105+
Liu, 2022 [29]	*Human* ADSC-sEVs	Ultracentrifugation	*Human* ADSCs: flow cytometry *human* ADSC-sEVs: TEM, NTA, nanoflow cytometry	~130 nm	*Human* ADSCs: CD73+, CD90+, CD105+, CD14−, CD19−, CD34−, CD45−, HLA-DR- *Human* ADSC-sEVs: CD9+, CD81+
Chen, 2013 [30]	*Rabbit* BMSCs	Centrifugation, adherence to tissue culture flask	ND	NA	ND
Lu, 2015 [31]	GFP-labeled *mouse* BMSCs	NR	NR	NA	ND
Zaki, 2017 [32]	*Rabbit* BMSCs	Centrifugation, adherence to tissue culture flask	Flow cytometry	NA	CD90+, CD105+, CD106+, CD45−
Zhang, 2017 [33]	GFP-labeled *mouse* BMSCs	NR	NR	NA	ND
Kim, 2019 [34]	*Human* UCMSCs	Density gradient centrifugation, adherence to tissue culture flask	Flow cytometry, proliferation, chondrogenic differentiation, HE, AB staining, gene expression analysis	NA	CD34−, CD45−, CD90+, CD105+ FGF-2, TGF-β1, IGF-1, Col1α1, Col2α1, ACAN, OCT4, NANOG, Sox2
Zhang, 2019 [35]	*Human* ESC-MSC exosomes	Size fractionation	WB, NTA, protein concentration	100–200 nm	CD81, ALIX, TSG101
Ogasawara, 2020 [36]	*Human* SHED-CM	Supernatants after centrifugation	LC-MS/MS analysis	NA	SPON2, IGF2, SDC4, SDC1, SFRP1, PTN, MDK, TGFb2, PDGFD, HGF
Cui, 2020 [37]	*Human* DPSCs	Centrifugation, adherence to tissue culture flask	NR	NA	NR
Wang, 2021 [38]	*Human* BMSC-derived sEVs	Centrifugation, microfiltration, ultrafiltration	NTA, TEM, immunoblotting	~106 nm	CD81+, CD63+, Rab5+, ALIX+, GRP94−
Köhnke, 2021 [39]	*Human* adipose-derived MSCs	NR	Flow cytometry, adipogenic, chondrogenic and osteogenic differentiation	NA	CD13+, CD44+, CD73+, CD90+, CD105+, CD31−, CD45−, CD235a−, HLA-II−
AbuBakr, 2022 [40]	*Rat* BMSC-derived MVs	Ultracentrifugation	TEM, FACS, ELISA	~79 nm	BMSCs: CD90+, CD105+, CD14− BMSC-derived MVs: CD63+, CD81+
Yang, 2022 [41]	*Rat* BMSCs	Centrifugation, adherence to tissue culture flask	SEM, immunocytochemical staining, ELISA, gene expression analysis	NA	Sox9, Col2a1, ACAN

AB, alcian blue; ACAN, aggrecan; ADRCs, adipose-derived regenerative cells; ADSC, adipose-derived stem cell; AE, ADSC-derived sEV; ALIX, ALG2-interacting protein; ALP, alkaline phosphatase; ARS, Alizarin Red S; BMSCs, bone marrow-derived mesenchymal stem cells; CLSM, confocal laser scanning microscopy; CM, conditioned media; SHED, *human* exfoliated deciduous teeth stem cells; DPSCs, dental pulp stem cells; ELISA, enzyme-linked immunosorbent assay; ESC, embryonic stem cell; FACS, fluorescence-activated cell sorting; FGF-2, fibroblast growth factor-2; GFP, green fluorescence protein; GRP94, glucose-regulated protein 94; HGF, hepatocyte growth factor; HIF-1α, hypoxia-inducible factor-1-alpha; HLA-DR, MHC class II cell surface receptor encoded by the *human* leukocyte antigen complex; IGF-1, insulin-like growth factor-1 IHC, immunohistochemistry; LC-MS/MS analysis, liquid chromatography tandem mass spectrometry; MDK, midkine; MSC, mesenchymal stem/stromal cell; MVs, microvesicles; NA, not applicable; NANOG, nanog homeobox; ND, not done; NR, not reported; NTA, nanoparticle tracking analysis; OCT, octamer-binding transcription factor; PDGF, platelet derived growth factor; PTN, pleiotrophin; Rab5, ras-related protein 5; SEM, scanning electron microscopy; sEVs, small extracellular vesicles; SDC, syndecan; SFRP, secreted frizzled related protein; Sox, sex-determining region Y-box; SPON, R-Spondin; TEM, transmission electron microscopy; TGF-β, transforming growth factor-beta; TSG101, tumor susceptibility gene 101; UCMSCs, umbilical cord mesenchymal stem cells; WB, Western blot.

**Table 3 cells-13-00990-t003:** Treatment parameters and analyses for included studies evaluating MSC-based therapies for cartilage/osteochondral defects.

Author, Year	Groups	Study Timepoints	Conc.	Volume/ Frequency	Delivery	Method of Analysis					
						Gross	Imaging	Histology	IHC	Molecular	Pain
Zhu, 2011 [19]	PLGA scaffold with NELL-1-modified BMSCs group, BMSCs group, PLGA group, empty defect group	6, 24 weeks	7.5 × 10^7^ cells/mL	40 μL/ one time	Implantation/PLGA scaffold	Gross morphology	Micro-CT (BV/TV)	SO, histological scoring	Col II	ND	ND
Sun, 2018 [20]	Gel-cell group, gel group	4, 8, 12 weeks	5 × 10^7^ cells/mL	NR/ one time	Implantation/Pluronic F-127 gel	Gross morphology	X-ray	HE, Wakitani scoring	Col II	ND	ND
Putnova, 2019 [21]	*Human* ADRC group, control group	11, 28 days	NR	1 mL/ one time	Flushing/ ADRC solution	Gross morphology	ND	HE, AB, Histomorphometry	ND	ND	ND
Cheng, 2019 [22]	Blank control group, PRF group, BMSC sheet group, BMSC/PRF construct group, pressure-pretreated BMSC/PRF construct group	2, 4, 8 weeks	NR	NR/ one time	Implantation/PRF scaffold	Gross morphology	ND	HE, TB, histological scoring	ND	*Sox9*, *Col II*, *ACAN*	ND
Sumarta, 2021 [23]	Untreated defects, defects treated with UCMSCs, defects treated with PRF scaffold, defects treated with UCMSCs and PRF scaffold	6 weeks	2 × 10^6^ cells	NR/ one time	Implantation/PRF	ND	ND	HE	Ki67, FGF18, Sox9, Col II, ACAN	ND	ND
Cheng, 2021 [24]	Empty group, collagen scaffold group, collagen scaffold with BMSCs group, collagen scaffold with HIF-1α overexpressing BMSCs group, sham group	12 weeks	2 × 10^5^ cells/mL	NR/ one time	Implantation/*rat* tail collagen scaffold	Gross morphology	Micro-CT (BV/TV, Tb.Th, Tb.Sp, Tb.N)	SO/FG, histological scoring	Col II, HIF-1α	ND	ND
Yu, 2021 [25]	Healthy control, defect group, bi-layered scaffold group, bi-layered scaffold with induced 14-day BMSCs group	2 months	5 × 10^7^ cells/mL	200 μL/ one time	Implantation/PEG crosslinked-PLGA-g-PCL scaffold	Gross morphology	ND	HE, SO/FG	Col I, Col II	ND	ND
Wang, 2021 [26]	Empty defect group, bi-layer scaffold group, bi-layer scaffold with BMSCs/chondrocyte group	6, 24 weeks	5 × 10^6^ cells/mL	NR/ one time	Implantation/HA-SH-Col I hydrogel-BCP ceramic scaffold	Gross morphology	Micro-CT (BMD, BV/TV, Tb.Th, Tb.N)	HE, Masson, SO/FG, O’Driscoll scoring	Col I, Col II, Col X	ND	ND
Guastaldi, 2021 [27]	Sham group, defect group, defect treated with MSCs + hydrogel + biosilica group	4, 8 weeks	1 × 10^6^ cells/mL	20 μL/ one time	Implantation/gelatine/biosilica-based hydrogel scaffold	ND	ND	HE, SO/FG	ND	ND	ND
Gomez, 2021 [28]	BMSC/PRP treated group, control group (untreated), sham group	6 weeks	10^5^ cells/mL	100 µL/ one time	Implantation/PRP	ND	ND	HE, AB, Mankin scoring	ND	ND	ND
Liu, 2022 [29]	Non-implanted group, scaffold group, scaffold with AE group, scaffold with IAE group	4, 8 weeks	5 µg sEV/scaffold	NR/ one time	Implantation/SF scaffold	Gross morphology (ICRS macro-scopic scoring)	Micro-CT (BV/TV, Tb.Th, Tb.Sp, Tb.N)	HE, SO/FG	Col I, Col II	ND	ND

AB, alcian blue; AE, ADSC-derived sEV; BCP, biphasic calcium phosphate; BMD, bone mineral density; BV/TV, bone volume over total volume; Conc., concentration; Col, collagen; FG, fast green; HA, hyaluronic acid; HIF-1α, HA-SH, self-crosslinking thiolated hyaluronic acid; hypoxia-inducible factor-1-alpha; HE, hematoxylin and eosin; IAE, inflammation-stimulated ADSC-derived sEV; ICRS, International Cartilage Regeneration & Joint Preservation Society; IHC, immunohistochemistry; Ki-67, Kiel 67; Micro-CT, micro-computed tomography; NELL-1, NEL-like protein-1; PCL, polycaprolactone; PEG, polyethylene glycol; PLGA, poly(L-glutamic acid); PRF, platelet-rich fibrin; PRP, platelet-rich plasma; ND, not done; NR, not reported; SO, safranin O; SF, silk fibroin; TB, toluidine blue; Tb.N, trabecular number; Tb.Th, trabecular thickness; Tb.Sp, trabecular spacing; TB, toluidine blue.

**Table 4 cells-13-00990-t004:** Treatment parameters and analyses for included studies evaluating the efficacy of MSC-based therapies for OA treatment.

Author, Year	Groups	Study Timepoints	Conc.	Volume/ Frequency	Delivery	Method of Analysis					
						Gross	Imaging	Histology	IHC	Molecular	Pain
Chen, 2013 [30]	Non-chondrogenic MSCs group, chondrogenic differentiated MSCs group, vehicle (hylartin) group, normal control group	4, 12, 24 weeks	2 × 10^6^ cells/mL	100 µL/ one time	I.A. injection/ hylartin solution	ND	Micro-CT (BV/TV, Tb.Th, Tb.Sp)	SO, TB, histological grading	Col II	*Sox9*, *Col2a1, ACAN*, *MMP13*	ND
Lu, 2015 [31]	Control group, BMSCs group, UAC group, UAC with BMSCs group	4, 8, 12 weeks	10^5^ cells/mL	20 µL/ weekly	I.A. injection/DMEM with 10% FCS	ND	Micro-CT (BV/TV, Tb.Th, Tb.Sp, Tb.N)	HE, SO, TRAP	Col II, SDF-1, RANTES, GFP	*Col1a1,2a1,10a1*, *ACAN*, *OCN*, *MMP13*, *TNF-α*, *IL-1β*	ND
Zaki, 2017 [32]	Untreated group, PBS group, BMSCs group, Arthritis with PBS group, Arthritis with BMSCs group	3 weeks	NR	200 µL/ one time	I.A. injection/ PBS	ND	ND	HE, SO, histomorphometry	ND	ND	ND
Zhang, 2017 [33]	Control group, UAC group, UAC with BMSCs group	4, 8, 12 weeks	10^5^ cells/mL	20 µL/ weekly	I.A. injection/ DMEM with 10% FCS	ND	TEM	HE, SO	Col I, Col II, Ki67, DAP3, CD163, GFP	ND	ND
Kim, 2019 [34]	Healthy control group; TMJ-OA group; TMJ-OA treated with DEX group; TMJ-OA treated with low, medium, or high dose of *human* UCMSCs groups	8 weeks	5 × 10^5^, 2.5 × 10^6^, 5 × 10^6^ cells/mL	200 µL/ one time	I.A. injection/ saline	ND	Micro-CT	HE, SO/FG, Mankin scoring	ACAN, Col I	*TNF-α*, *IL-1β,-6,-10,-17*, *TGF-β1*, *IGF-1*, *FGF-2*, *Col1a1,2a1*, *ACAN*	ND
Zhang, 2019 [35]	OA + PBS group, OA + exosome group, sham group	2, 4, 8 weeks	2 mg/mL	50 µL/ weekly	I.A. injection/ PBS	ND	Micro-CT (BV/TV Tb.Th Tb.Sp Tb.N)	HE, TB, Mankin scoring, histomorphometry	α-SMA, MMP13, IL-1β, iNOS, PCNA, CCP3	*IL-1β,-4,-10 iNOS*, *TNF-α*, *TGF-β1*, *Col1a1,2a1*, *Sox9*, *COMP*, *ACAN*, *MMP3,9,13* *TIMP1,2,3*, *ADAMTS4,5*, *BAX*, *Casp3,8,9*, *Survivin*, *PCNA*, *α-SMA*, *SP*, *CGRP*, *NGF*, *P75NTR*, *TrkA*	HWT
Ogasawara, 2020 [36]	Sham group, pre-treatment group, DMEM group, SHED-CM group,	12 days	3 μg/mL	0.5 mL/ 5 days	Tail vein injection/ serum-free DMEM	ND	Micro-CT (BV/TV, Tb.Th, Tb.Sp)	HE, TB, Mankin scoring, TUNEL, TRAP	MMP13, iNOS, IL-1β, PCNA	ND	HWT
Cui, 2020 [37]	Control group, TMJ-OA with saline group, TMJ-OA with DPSCs group	2, 4 weeks	4 × 10^6^ cells/mL	50 µL/ one time	I.A. injection/ saline	ND	Micro-CT (BV/TV BS/BV Tb.Sp Tb.N)	HE, TB, Mankin scoring	CD4, IFN-γ, TNF-α, MMP3, MMP13	ND	HWT
Wang, 2021 [38]	Control group, OA group, sEV group, OA + sEV group	4, 6, 8 weeks	(2–4) × 10^9^ particles/mL	200 µL/ one time	I.A. injection/ PBS	Gross morphology	Micro-CT (BV/TV BS/BV BMD Tb.Th) Tb.Sp	HE, SO/FG, AB, Wakitani scoring	PCNA, Col I, Col II, ACAN, Sox9, MMP13, RUNX2	ND	ND
Köhnke, 2021 [39]	AB serum group, HA group, MSCs group, MSCs + HA group	4 weeks	10^6^ cells/mL	150 µL/ one time	I.A. injection/ HA	ND	CT, SEM	SO, Picrosirius red	ND	ND	ND
AbuBakr, 2022 [40]	OA group, OA + MVs group, OA + PRP group	2, 4 weeks.	2 mg/mL	50 µL/ one time	I.A. injection/ PBS	ND	ND	HE	ND	*IL-1β*, *TNF-α*, *NF-κB*, *MMP3,13*, *Col II*	ND
Yang, 2022 [41]	BMSC-coated microspheres group, microspheres group, control group	1, 2 weeks	NR	200 µL/ one time	I.A. injection/ GelMA microspheres	ND	SEM, Micro-CT (BMD, BV/TV)	HE	Sox9	ND	ND

ADAMTS, A disintegrin and metalloproteinase with thrombospondin motifs; AB, alcian blue; α-SMA, α-smooth muscle actin; BAX, bcl2 associated x; BMD, bone mineral density; BS/BV; bone surface over bone volume; BV/TV; bone volume over total volume; CCP3, cleaved caspase-3; CGRP, calcitonin gene-related peptide; COMP, cartilage oligomeric matrix protein; CT, computed tomography; DAP3, death-associated protein 3; DEX, dexamethasone; DMEM, Dulbecco’s Modified Eagle Medium; FG, fast green; FCS, fetal calf serum; FGF-2, fibroblast growth factor-2; GelMA, gelatin methacryloyl; HA, hyaluronic acid; HE, hematoxylin and eosin; HWT, head withdrawal threshold; I.A., intra-articular; IFN- γ, interferon-γ; IGF, insulin-like growth factor-1; IL-1β, interleukin-1beta; iNOS, inducible nitric oxide synthase; MMP, matrix metalloproteinase; ND, not done; NF-κB, nuclear factor kappa B; NGF, nerve growth factor; OCN, osteocalcin; P75NTR, p75 neurotrophin receptor; PCNA, proliferative cell nuclear antigen; RANTES, regulated on activation normal T-cell expressed and secreted; RUNX2, RUNX family transcription factor 2; SDF-1, stromal cell-derived factor 1; SO, safranin O; SP, substance P; TB, toluidine blue; Tb.N, trabecular number; Tb.Th, trabecular thickness; Tb.Sp, trabecular spacing; TGF-β1, transforming growth factor-beta-1; TIMP, tissue inhibitor metalloproteinase; TNF-α, tumor necrosis factor-alpha; TRAP, tartrate-resistant acid phosphatase; TrkA, tropomyosin receptor kinase A; TUNEL, terminal deoxynucleotidyl transferase (TdT) dUTP nick-end labeling.

**Table 5 cells-13-00990-t005:** Summary of key outcomes of included studies evaluating MSC-based therapies for cartilage/osteochondral defects.

Author, Year	Key Outcomes (In Vitro)	Key Outcomes (In Vivo)
Zhu, 2011 [19]	ND	NELL-1-modified BMSCs/PLGA composite rapidly repaired large osteochondral defect in the mandibular condyle with regeneration of fibrocartilage and subchondral bone.
Sun, 2018 [20]	ND	Co-culture of *goat* BMSCs and chondrocytes at 7:3 ratio in hydrogel induced chondrogenic differentiation of BMSCs to enhance TMJ repair.
Putnova, 2019 [21]	ND	*Human* ADRCs supported soft tissue repair and promoted bone remodeling in hard tissues.
Cheng, 2019 [22]	Hydrostatic pressure pre-treatment (120 kPa/1 h for 4 days) optimally promoted BMSC proliferation and chondrogenic differentiation in the BMSC/PRF construct.	Pressure-pretreated BMSC/PRF construct enhanced cartilage regeneration with improved mechanical properties and integration of the neocartilage.
Sumarta, 2021 [23]	ND	*Human* UCMSCs in PRF scaffold proved capable of regenerating mandibular cartilage defect through increased expression of *FGF-18*, *Sox9*, *Ki67*, *Col II*, *ACAN*, and cartilage thickness
Cheng, 2021 [24]	ND	Transplantation of HIF-1α overexpressed BMSCs combined with a collagen scaffold promoted cartilaginous repair of condylar cartilage and inhibited subchondral bone sclerosis in TMJ osteochondral defect.
Yu, 2021 [25]	The bi-layered PLGA-g-PCL scaffold with segregated hydrophilicity–hydrophobicity facilitated chondrogenic differentiation of BMSCs toward top fibrocartilage layer and bottom hyaline cartilage layer.	The bi-layered PLGA-g-PCL-PEG scaffold with segregated hydrophilicity–hydrophobicity carrying induced 14-day BMSCs enabled reconstruction of *goat* hierarchical TMJ condylar cartilage.
Wang, 2021 [26]	The HA-SH/Col I blend hydrogel-BCP ceramic bi-layered scaffold enhanced proliferation and matrix synthesis of *rabbit* BMSCs and chondrocytes, as well as osteogenic differentiation of BMSCs.	*Rabbit* BMSCs/chondrocytes-loaded bi-layer scaffold could effectively promote the regeneration of both fibrocartilage and subchondral bone.
Guastaldi, 2021 [27]	ND	The MSCs + hydrogel + biosilica was effective in promoting TMJ condylar cartilage regeneration, as evidenced by intact articular surfaces, maturation, and distribution of chondrocytes along the condyle.
Gomez, 2021 [28]	ND	BMSC/PRP implantation promoted repair of the articular surface with the presence of cartilage-like tissue and subchondral bone filling the defect area.
Liu, 2022 [29]	Both IAE and AE showed comparable effects on proliferation, whereas IAE outperformed AE on BMSC migration and M2 macrophage polarization in vitro. RNA sequencing identified high miR-27b-3p expression levels in IAE that may regulate macrophage polarization by targeting CSF-1.	IAE loaded onto SF scaffold outperformed AE-loaded scaffold in TMJ osteochondral regeneration, with newly formed cartilage stained for GAG, collagen I and II, and reconstituted subchondral bone.

AE, ADSC-derived sEV; BMSCs, bone marrow-derived mesenchymal stem cells; CSF-1, colony-stimulating factor-1; GAG, glycosaminoglycan; HA-SH, self-crosslinking thiolated hyaluronic acid; HIF-1α, hypoxia-inducible factor-1-alpha; ND, not done; NELL-1, NEL-like protein-1; PCL, polycaprolactone; PEG, polyethylene glycol; PLGA, poly-lactic-co-glycolic acid; PLGA-g-PCL-PEG, poly(L-glutamic acid)-graft-poly(Ɛ-caprolactone)-poly(ethylene glycol); PRF, platelet-rich fibrin; PRP, platelet-rich plasma; IAE, inflammation-stimulated ADSC-derived sEV; SF, silk fibroin.

**Table 6 cells-13-00990-t006:** Summary of key outcomes for included studies evaluating MSC-based therapies for OA treatment.

Author, Year	Key Outcomes (In Vitro)	Key Outcomes (In Vivo)
Chen, 2013 [30]	ND	I.A injection of MSCs could delay the progression of TMJ-OA, and in vitro chondrogenic differentiation of MSCs could enhance the therapeutic effects.
Lu, 2015 [31]	SDF-1 and RANTES were significantly increased in the UAC cartilage compared to the control cartilage. Migration of BMSCs was enhanced when cocultured with a UAC TMJ condyle and was attenuated in the presence of AMD3100 (CXCR4 antagonist) or BX471 (CCR1 antagonist).	BMSC injections improved cartilage repair and subchondral bone restoration in TMJ-OA *mice* induced by UAC. The locally injected BMSCs were found to implant and differentiate into chondrocytes in OA cartilage. These effects of BMSCs were inhibited by AMD3100 and BX471.
Zaki, 2017 [32]	ND	*Rabbit* BMSCs could safely and effectively repair degenerative changes of *rabbit* TMJs with bovine collagen II-induced arthritis.
Zhang, 2017 [33]	Fluid flow shear stress (FFSS) stimulation induced cell death of superficial and deep zone chondrocytes. Genes associated with chondrocyte hypertrophy and fibrosis were upregulated in deep zone chondrocytes with FFSS stimulation.	BMSCs rescued the damaged cartilage by increasing matrix production and scavenging activity.
Kim, 2019 [34]	*Human* UCMSC lines isolated from different donors showed comparable proliferation ability but varying in vitro capacities for chondrogenesis and expression of marker genes for growth factors and ECM compared to that of BMSCs.	*Human* UCMSCs exerted anti-inflammatory effects and promoted cartilage regeneration in a *rabbit* model of TMJ-OA. Medium dose of MSCs was most effective in regenerating the articular cartilage with the highest gene expression levels of growth factors.
Zhang, 2019 [35]	*Human* ESC-MSC exosomes suppressed inflammation and restored matrix synthesis in IL-1β-treated chondrocytes via adenosine receptor activation of AKT, ERK, and AMPK pathways.	*Human* ESC-MSC exosomes suppressed pain and inflammation and reduced cell apoptosis and matrix degradation while enhancing cell proliferation and matrix synthesis to promote overall TMJ repair and regeneration.
Ogasawara, 2020 [36]	LC-MS/MS analysis identified several factors present in SHED-CM that could be involved in processes such as anti-fibrosis, anti-apoptosis, anti-inflammation, proliferation, differentiation, and migration of chondrocytes.	SHED-CM contained multiple therapeutic factors with the potential to promote the regeneration and repair of mechanical-stress-induced *mouse* TMJ-OA.
Cui, 2020 [37]	DPSCs downregulated the expression of MMP3 and MMP13 in fibroblast-like synoviocytes by suppressing STAT1 activation under the inflammatory condition.	DPSC local injection relieved pain, suppressed synovial inflammation, and reduced cartilage degradation and subchondral bone destruction in *rats*.
Wang, 2021 [38]	BMSC-sEVs enhanced proliferation and migration of mandibular condylar chondrocytes, possibly through activation of the Hippo pathway.	BMSC-sEVs promoted cartilage reconstruction in TMJ-OA via the autotaxin–YAP signaling axis.
Köhnke, 2021 [39]	ND	*Human* adipose-derived MSCs with or without HA were more effective than serum and HA alone in restoring the cartilage thickness in a *rabbit* TMJ-OA model.
AbuBakr. 2022 [40]	ND	BMSC-derived MVs restored damaged condylar structure by suppressing inflammation and matrix degradation in a *rat* model of TMJ-OA.
Yang, 2022 [41]	GelMA microspheres loaded with TGF-β enhanced chondrogenic differentiation of BMSCs coated on the microspheres.	*Rat* BMSC-coated GelMA microspheres endowed with superwettable properties and sustained TGF-β release, can efficiently colonize the bone defect site, release cytokine, and promote cartilage healing.

BMSCs, bone marrow-derived mesenchymal stem cells; CCR, C-C chemokine receptor; CXCR, C-X-C chemokine receptor; DPSCs, dental pulp stem cells; GelMA, gelatin methacryloyl; ESC, embryonic stem cell; ECM, extracellular matrix; FFSS, fluid flow shear stress; HA, hyaluronic acid; I.A., intra-articular; MVs, microvesicles; ND, not done; RANTES, regulated on activation normal T-cell expressed and secreted; SDF-1, stromal cell-derived factor 1; SHED-CM, conditioned media from *human* exfoliated deciduous teeth stem cells; TMJ-OA, temporomandibular joint osteoarthritis; TGF-β; transforming growth factor-beta; UAC, unilateral anterior cross-bite; UCMSCs, umbilical cord mesenchymal stem cells; YAP, yes-associated protein.

**Table 7 cells-13-00990-t007:** ARRIVE checklist to evaluate the quality of the included studies for the repair of cartilage/osteochondral defects.

Item/Item Number	Zhu, 2011 [19]	Sun, 2018 [20]	Putnova, 2019 [21]	Cheng, 2019 [22]	Sumarta, 2021 [23]	Cheng, 2021 [24]	Yu, 2021 [25]	Wang, 2021 [26]	Guastaldi, 2021 [27]	Gomez, 2021 [28]	Liu, 2022 [29]	Cat. Score	Total Score	Ratio
1. Title (0, inaccurate/not concise; 1, accurate/concise)	1	1	1	1	1	1	1	1	1	1	1	11	11	1
2. Abstract Summary of the background; research objectives, including details of the species or strain of animal used; key methods; principal findings; and conclusions of the study (0, clearly inaccurate; 1, possibly accurate; 2, clearly accurate)	2	1	1	2	2	2	1	1	2	2	2	18	22	0.82
3. Introduction Background: objectives, experimental approach and rationale, and relevance to human biology (0, clearly insufficient; 1, possibly sufficient; 2, clearly sufficient)	1	1	2	1	1	1	1	1	1	1	2	13	22	0.59
4. Introduction Objectives: primary and secondary (0, not clear; 1, clear)	1	1	1	1	0	0	1	1	0	0	1	7	11	0.64
5. Methods Ethical statement: nature of the review permission, relevant licenses, and national and institutional guidelines for the care and use of animals (0, clearly insufficient; 1, possibly sufficient; 2, clearly sufficient)	1	1	2	2	2	1	1	1	2	2	1	16	22	0.73
6. Methods Study design: number of experimental and control groups and any steps taken to minimize bias (i.e., allocation concealment, randomization, blinding) (0, clearly insufficient; 1, possibly sufficient; 2, clearly sufficient)	1	2	2	2	1	2	2	2	1	0	0	15	22	0.68
7. Methods Experimental procedure: precise details (i.e., how, when, where, why) (0, clearly insufficient; 1, possibly sufficient; 2, clearly sufficient)	1	1	2	1	1	1	2	1	2	2	1	15	22	0.68
8. Methods Experimental animals: species, strain, sex, developmental stage, weight, and source of animals (0, clearly insufficient; 1, possibly sufficient; 2, clearly sufficient)	1	1	1	1	1	1	1	1	1	1	1	11	22	0.5
9. Methods Housing and husbandry: conditions and welfare-related assessments and interventions (0, clearly insufficient; 1, possibly sufficient; 2, clearly sufficient)	0	1	2	1	0	0	1	1	1	2	0	9	22	0.41
10. Methods Sample size: total number of animals used in each experimental group, details of calculation (0, clearly inadequate; 1, possibly inadequate; 2, clearly adequate)	1	2	1	1	1	1	1	1	2	1	0	12	22	0.55
11. Methods Allocation animals to experimental groups: randomization or matching, order in which animals were treated and assessed (0, no; 1, yes)	0	1	1	1	1	1	1	1	0	0	0	7	11	0.64
12. Methods Experimental outcomes: definition of primary and secondary outcomes (0, no; 1, unclear/not complete; 2, yes)	0	0	0	0	0	0	0	0	0	0	0	0	22	0
13. Methods Statistical methods: details and unit of analysis (0, no; 1, unclear/ not complete; 2, yes)	2	2	1	2	1	2	2	2	0	0	2	16	22	0.73
14. Results Baseline data: characteristics and health status of animals (0, no; 1, yes)	0	0	0	1	0	1	1	1	1	0	0	5	11	0.45
15. Results Numbers analyzed: absolute numbers in each group included in each analysis, explanation for exclusion (0, clearly inadequate; 1, possibly inadequate; 2, clearly adequate)	1	0	0	1	0	1	1	1	2	0	1	8	22	0.36
16. Results Outcomes and estimation: results for each analysis with a measure of precision (0, no; 1, unclear/not complete; 2, yes)	2	1	1	1	2	1	1	1	1	1	2	14	22	0.64
17. Results Adverse events: details and modifications for reduction (0, no; 1, unclear/not complete; 2, yes)	0	0	0	0	0	0	0	0	2	1	0	3	22	0.14
18. Discussion Interpretation/scientific implications: study limitations including animal model, and implications for the 3Rs (0, clearly inadequate; 1, possibly inadequate; 2, clearly adequate)	1	1	0	1	0	1	1	2	1	1	1	10	22	0.45
19. Discussion Generalizability/translation: relevance to human biology (0, clearly inadequate; 1, possibly inadequate; 2, clearly adequate)	2	2	2	2	1	1	1	2	1	1	2	17	22	0.77
20. Discussion Funding: resources and role of the funders (0, clearly inadequate; 1, possibly inadequate; 2, clearly adequate)	2	2	2	2	1	2	2	2	2	2	2	21	22	0.95

**Table 8 cells-13-00990-t008:** ARRIVE checklist evaluating the quality of the included studies for OA treatment.

Item/Item Number	Chen, 2013 [30]	Lu, 2015 [31]	Zaki, 2017 [32]	Zhang, 2017 [33]	Kim, 2019 [34]	Zhang, 2019 [35]	Ogasawara, 2020 [36]	Cui, 2020 [37]	Wang, 2021 [38]	Köhnke, 2021 [39]	AbuBakr, 2022 [40]	Yang, 2022 [41]	Cat. Score	Total Score	Ratio
1. Title (0, inaccurate/not concise; 1, accurate/concise)	1	1	1	1	1	1	1	1	1	1	1	1	12	12	1
2. Abstract Summary of the background; research objectives, including details of the species or strain of animal used; key methods; principal findings; and conclusions of the study (0, clearly inaccurate; 1, possibly accurate; 2, clearly accurate)	1	1	1	2	1	1	1	1	1	1	2	1	14	24	0.58
3. Introduction Background: objectives, experimental approach and rationale, and relevance to human biology (0, clearly insufficient; 1, possibly sufficient; 2, clearly sufficient)	1	1	1	2	1	2	1	1	1	1	1	1	14	24	0.58
4. Introduction Objectives: primary and secondary (0, not clear; 1, clear)	0	1	0	0	0	1	0	1	1	0	0	1	5	12	0.42
5. Methods Ethical statement: nature of the review permission, relevant licenses, and national and institutional guidelines for the care and use of animals (0, clearly insufficient; 1, possibly sufficient; 2, clearly sufficient)	1	1	1	1	2	2	2	2	2	2	2	2	20	24	0.83
6. Methods Study design: number of experimental and control groups and any steps taken to minimize bias (i.e., allocation concealment, randomization, blinding) (0, clearly insufficient; 1, possibly sufficient; 2, clearly sufficient)	1	0	2	0	2	2	2	2	2	2	2	2	19	24	0.79
7. Methods Experimental procedure: precise details (i.e., how, when, where, why) (0, clearly insufficient; 1, possibly sufficient; 2, clearly sufficient)	2	1	1	1	1	2	1	1	1	2	2	1	16	24	0.67
8. Methods Experimental animals: species, strain, sex, developmental stage, weight, and source of animals (0, clearly insufficient; 1, possibly sufficient; 2, clearly sufficient)	1	2	1	2	1	1	1	1	1	1	1	1	14	24	0.58
9. Methods Housing and husbandry: conditions and welfare-related assessments and interventions (0, clearly insufficient; 1, possibly sufficient; 2, clearly sufficient)	1	0	1	1	0	2	1	1	0	2	2	0	11	24	0.46
10. Methods Sample size: total number of animals used in each experimental group and details of calculation (0, clearly inadequate; 1, possibly inadequate; 2, clearly adequate)	2	0	2	0	2	1	2	1	1	0	1	0	12	24	0.5
11. Methods Allocation of animals to experimental groups: randomization or matching and order in which animals were treated and assessed (0, no; 1, yes)	0	0	1	0	1	1	1	1	1	1	1	1	9	12	0.75
12. Methods Experimental outcomes: definition of primary and secondary outcomes (0, no; 1, unclear/not complete; 2, yes)	0	0	0	0	0	0	0	0	0	0	0	0	0	24	0
13. Methods Statistical methods: details and unit of analysis (0, no; 1, unclear/ not complete; 2, yes)	2	2	2	2	2	2	1	1	2	1	2	2	21	24	0.88
14. Results Baseline data: characteristics and health status of animals (0, no; 1, yes)	1	1	1	0	1	1	1	1	0	1	0	0	8	12	0.67
15. Results Numbers analyzed: absolute numbers in each group included in each analysis and explanation for exclusion (0, clearly inadequate; 1, possibly inadequate; 2, clearly adequate)	0	1	0	0	1	2	1	1	1	0	0	1	8	24	0.33
16. Results Outcomes and estimation: results of each analysis with a measure of precision (0, no; 1, unclear/not complete; 2, yes)	1	1	2	1	1	2	2	1	1	1	1	1	15	24	0.63
17. Results Adverse events: details and modifications for reduction (0, no; 1, unclear/not complete; 2, yes)	0	0	1	0	2	2	1	0	2	2	0	0	10	24	0.42
18. Discussion Interpretation/scientific implications: study limitations including animal model, and implications for the 3Rs (0, clearly inadequate; 1, possibly inadequate; 2, clearly adequate)	1	1	1	1	1	1	1	1	1	1	1	0	11	24	0.46
19. Discussion Generalizability/translation: relevance to human biology (0, clearly inadequate; 1, possibly inadequate; 2, clearly adequate)	1	1	1	1	2	2	1	2	2	2	1	1	17	24	0.71
20. Discussion Funding: resources and role of the funders (0, clearly inadequate; 1, possibly inadequate; 2, clearly adequate)	2	2	0	2	2	2	2	2	2	2	2	2	22	24	0.92

**Table 9 cells-13-00990-t009:** Risk of bias assessed using SYRCLE risk of bias assessment tool.

Author, Year	Selection Bias			Performance Bias		Detection Bias						Attrition Bias	Reporting Bias	Other
	Sequence Generation	Baseline Characte-ristics	Allocation Conceal-ment	Random Housing	Blinding	Random Outcome Assessment	Gross Blinding	Imaging Blinding	Histology Blinding	IHC Blinding	Molecular Blinding	Incomplete Outcome Data	Selective Outcome Reporting	Other Bias
Zhu, 2011 [19]	Unclear Risk	Unclear Risk	Unclear Risk	Unclear Risk	Unclear Risk	Unclear Risk	Unclear Risk	Unclear Risk	Low Risk	Unclear Risk	ND	Unclear Risk	Low Risk	Low Risk
Sun, 2018 [20]	Unclear Risk	Unclear Risk	Low Risk	Unclear Risk	Unclear Risk	Unclear Risk	Unclear Risk	Unclear Risk	Unclear Risk	Unclear Risk	ND	Unclear Risk	Low Risk	Low Risk
Putnova, 2019 [21]	Unclear Risk	Low Risk	Low Risk	Unclear Risk	Unclear Risk	Unclear Risk	Unclear Risk	ND	Unclear Risk	ND	ND	Unclear Risk	Low Risk	Low Risk
Cheng, 2019 [22]	Unclear Risk	Unclear Risk	Low Risk	Unclear Risk	Unclear Risk	Low Risk	Unclear Risk	ND	Low Risk	ND	Unclear Risk	Unclear Risk	Low Risk	Low Risk
Sumarta, 2021 [23]	Unclear Risk	Low Risk	Low Risk	Unclear Risk	Unclear Risk	Unclear Risk	ND	ND	Unclear Risk	Unclear Risk	ND	Unclear Risk	Low Risk	Low Risk
Cheng, 2021 [24]	Unclear Risk	Low Risk	Low Risk	Unclear Risk	Unclear Risk	Low Risk	Unclear Risk	Unclear Risk	Unclear Risk	Unclear Risk	ND	Unclear Risk	Low Risk	Low Risk
Yu, 2021 [25]	Unclear Risk	Low Risk	Low Risk	Unclear Risk	Unclear Risk	Unclear Risk	Unclear Risk	ND	Unclear Risk	Unclear Risk	ND	Unclear Risk	Low Risk	Low Risk
Wang, 2021 [26]	Unclear Risk	Low Risk	Low Risk	Unclear Risk	Unclear Risk	Unclear Risk	Unclear Risk	Unclear Risk	Unclear Risk	Unclear Risk	ND	Unclear Risk	Low Risk	Low Risk
Guastaldi, 2021 [27]	Unclear Risk	Low Risk	Unclear Risk	Unclear Risk	Unclear Risk	Unclear Risk	ND	ND	Unclear Risk	ND	ND	Low Risk	Low Risk	Low Risk
Gomez, 2021 [28]	Unclear Risk	Low Risk	Unclear Risk	Unclear Risk	Unclear Risk	Unclear Risk	Unclear Risk	ND	Unclear Risk	ND	ND	Low Risk	Low Risk	Low Risk
Liu, 2022 [29]	Unclear Risk	Low Risk	Unclear Risk	Unclear Risk	Unclear Risk	Unclear Risk	Low Risk	Unclear Risk	Low Risk	Unclear Risk	ND	Unclear Risk	Low Risk	Low Risk
Chen, 2013 [30]	Unclear Risk	Low Risk	Unclear Risk	Unclear Risk	Unclear Risk	Low Risk	Unclear Risk	Unclear Risk	Low Risk	Unclear Risk	Unclear Risk	Unclear Risk	Low Risk	Low Risk
Lu, 2015 [31]	Unclear Risk	Low Risk	Unclear Risk	Unclear Risk	Unclear Risk	Unclear Risk	ND	Unclear Risk	Unclear Risk	Unclear Risk	Unclear Risk	Unclear Risk	Low Risk	Low Risk
Zaki, 2017 [32]	Unclear Risk	Low Risk	Low Risk	Unclear Risk	Unclear Risk	Unclear Risk	ND	ND	Unclear Risk	ND	ND	Low Risk	Low Risk	Low Risk
Zhang, 2017 [33]	Unclear Risk	Low Risk	Unclear Risk	Unclear Risk	Unclear Risk	Unclear Risk	ND	Unclear Risk	Unclear Risk	Unclear Risk	ND	Unclear Risk	Low Risk	Low Risk
Kim, 2019 [34]	Unclear Risk	Low Risk	Low Risk	Unclear Risk	Unclear Risk	Unclear Risk	ND	Unclear Risk	Low Risk	Unclear Risk	Unclear Risk	Low Risk	Low Risk	Low Risk
Zhang, 2019 [35]	Unclear Risk	Low Risk	Low Risk	Unclear Risk	Unclear Risk	Unclear Risk	ND	Unclear Risk	Low Risk	Unclear Risk	Unclear Risk	Low Risk	Low Risk	Low Risk
Ogasawara, 2020 [36]	Unclear Risk	Low Risk	Low Risk	Unclear Risk	Unclear Risk	Unclear Risk	ND	Unclear Risk	Unclear Risk	Unclear Risk	ND	Low Risk	Low Risk	Low Risk
Cui, 2020 [37]	Unclear Risk	Low Risk	Low Risk	Unclear Risk	Unclear Risk	Unclear Risk	ND	Unclear Risk	Low Risk	Unclear Risk	ND	Unclear Risk	Low Risk	Low Risk
Wang, 2021 [38]	Unclear Risk	Low Risk	Low Risk	Unclear Risk	Unclear Risk	Unclear Risk	Unclear Risk	Unclear Risk	Low Risk	Unclear Risk	ND	Low Risk	Low Risk	Low Risk
Köhnke, 2021 [39]	Unclear Risk	Low Risk	Low Risk	Unclear Risk	Unclear Risk	Unclear Risk	ND	Unclear Risk	Low Risk	ND	ND	Low Risk	Low Risk	Low Risk
AbuBakr, 2022 [40]	Unclear Risk	Low Risk	Low Risk	Unclear Risk	Unclear Risk	Unclear Risk	ND	ND	Unclear Risk	ND	Unclear Risk	Unclear Risk	Low Risk	Low Risk
Yang, 2022 [41]	Unclear Risk	Low Risk	Low Risk	Unclear Risk	Unclear Risk	Unclear Risk	ND	Unclear Risk	Unclear Risk	Unclear Risk	ND	Unclear Risk	Low Risk	Low Risk

IHC, immunohistochemistry; ND, not done.

## Data Availability

Data are available upon reasonable request to the corresponding authors.

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
