# Peer review of "Mesenchymal Stem Cell-Based Therapies for Temporomandibular Joint Repair: A Systematic Review of Preclinical Studies"

_cells, 2024, doi:10.3390/cells13110990_

Round 1

Reviewer 1 Report

Comments and Suggestions for Authors

In the systematic review entitled "Mesenchymal Stem Cell-Based Therapies for Temporomandibular Joint Repair: A Systematic Review of Preclinical Studies", the Authors reported the results of regenerative in vivo studies (until May 30, 2023) in which MSCs, their derived secretome, or EVs were used. They evaluated for each study the quality and risk of bias as wel as the presence of incomplete reporting data. The study is interesting and raises the issue about data standardization and reporting. However, some points should be clarified:

1. The Authors reported literature (the latest references are dated 2022) until May 30, 2023, so a year ago.  Have other works been published in the meantime?

2. In the Table 7 and 8, some scores were reported. Would it be possible to summarize giving a total score for each study and a general trend or threshold?

Reviewer 2 Report

Comments and Suggestions for Authors

The authors insisted that the principal finding of this systematic review is that MSC-based therapies are efficacious in treatment of TMJ cartilage/osteochondral defects and OA. This therapeutic efficacy of MSC-based therapies in TMJ repair is demonstrated by cartilage regeneration and subchondral bone restoration, with overall improvements in the morphological, histological, molecular and pain behavioral outcomes, in the studies reviewed. 

comments

1. In this review, it is not clear whether MSC, MSC-secretome, or MSC-EV is effective. It is also not clear which route is the best way to port. Which method is preferable in terms of cost, safety, and long-term effectiveness?

2. Is there any literature that states which of MSC, MSC-secretome, or MSC-EV is currently effective in human clinical research? Does it also mention the route of administration?

3. It seems necessary to make recommendations as to what kind of research is essential in the future in order to establish a TMD treatment method using MSCs.

Reviewer 3 Report

Comments and Suggestions for Authors

Some data in the table shows a lack of clarity regarding the primary and secondary objectives of the study. The authors are advised to clarify the tables.
It is also suggested that the authors add studies regarding the role of  mimetic peptides for regeneration and/or therapy. An example is given https://www.ncbi.nlm.nih.gov/pmc/articles/PMC9266819/

Comments on the Quality of English Language

minor revision
